# Plasma proteomic profiles predict individual future health risk

Jia You ®[1,7], Yu Guo[1,7], Yi Zhang[1,7], Ju-Jiao Kang ®[1], Lin-Bo Wang ®[1], Jian-Feng Feng ®[1,2,3,4,5] ✉, Wei Cheng ®[1,2,5,6] ✉ & Jin-Tai Yu ®[1] ✉

Developing a single-domain assay to identify individuals at high risk of future events is a priority for multi-disease and mortality prevention. By training a neural network, we developed a disease/mortality-specific proteomic risk score (ProRS) based on 1461 Olink plasma proteins measured in 52,006 UK Biobank participants. This integrative score markedly stratified the risk for 45 common conditions, including infectious, hematological, endocrine, psychiatric, neurological, sensory, circulatory, respiratory, digestive, cutaneous, musculoskeletal, and genitourinary diseases, cancers, and mortality. The discriminations witnessed high accuracies achieved by ProRS for 10 endpoints (e.g., cancer, dementia, and death), with C-indexes exceeding 0.80. Notably, ProRS produced much better or equivalent predictive performance than established clinical indicators for almost all endpoints. Incorporating clinical predictors with ProRS enhanced predictive power for most endpoints, but this combination only exhibited limited improvement when compared to ProRS alone. Some proteins, e.g., GDF15, exhibited important discriminative values for various diseases. We also showed that the good discriminative performance observed could be largely translated into practical clinical utility. Taken together, proteomic profiles may serve as a replacement for complex laboratory tests or clinical measures to refine the comprehensive risk assessments of multiple diseases and mortalities simultaneously. Our models were internally validated in the UK Biobank; thus, further independent external validations are necessary to confirm our findings before application in clinical settings.

Risk stratification is critical for the identification of high-risk individuals and disease prevention, especially at an early preclinical stage[1,2]. However, comprehensive risk assessments of human diseases frequently require a rigorous accumulation of predictors, one disease at a time. The resultant risk score for each disease would be severely restricted in clinical practical application because of the time and cost involved in gathering the information[3]. Accordingly, a single-domain assay that could inform on multiple diseases simultaneously becomes crucial[4]. As many nations now recommend routine check-ups entailing blood tests in the prevention of several common diseases, proteomics-based risk scores may hold a major promise to improve multi-disease risk prediction[5,6].

[1]Department of Neurology and National Center for Neurological Disorders, Huashan Hospital, Institute of Science and Technology for Brain-Inspired Intelligence, State Key Laboratory of Medical Neurobiology and MOE Frontiers Center for Brain Science, Fudan University, Shanghai, China. [2]Key Laboratory of Computational Neuroscience and Brain-Inspired Intelligence (Fudan University), Ministry of Education, Shanghai, China. [3]Zhangjiang Fudan International Innovation Center, Shanghai, China. [4]School of Data Science, Fudan University, Shanghai, China. [5]Fudan ISTBI—ZJNU Algorithm Centre for Brain-inspired Intelligence, Zhejiang Normal University, Zhejiang, China. [6]Shanghai Medical College and Zhongshan Hospital Immunotherapy Technology Transfer Center, Shanghai, China. [7]These authors contributed equally: Jia You, Yu Guo, Yi Zhang. ✉e-mail: jianfeng64@gmail.com; wcheng.fdu@gmail.com; jintai_yu@fudan.edu.cn

The human blood proteome provides a holistic readout of human health states through an untargeted assessment of thousands of circulating molecules, which can integrate the influences and interactions of genetics, lifestyle, environment, comorbidities, and drugs[7]. While this 'omics' approach has been linked to the discovery and understanding of gene-protein interactions[8–12], biomarkers in individual diseases and risks[5,13–18], aging[19–21], and drug pharmacology[22], its potential as a convenient tool to simultaneously and systematically assess the risk of multiple future health problems remains to be investigated. To date, most proteomic-based predictive studies have been undertaken in cross-sectional manners, and several of them leveraged case-control approaches to understand the plasma proteomic difference between healthy population and individuals diagnosed with certain diseases, e.g., dementia[23], Alzheimer's disease[24], coronary heart disease[25] and Type I diabetes[26]. Possibly biased by reverse causality, such studies have failed to identify proteomic signatures prior to disease onset. Although case-control studies are informative and can elaborate disease-related protein profiles to a certain extent, longitudinal designs that estimate early molecular signatures associated with disease incidence are more appropriate for risk stratification. Furthermore, despite the fact that the same molecular basis has been revealed in several closely linked diseases[27–29], knowledge of the shared pathways underlying less overtly relevant ones is sparse, and a correspondingly systematic understanding of different incident diseases in humans is lacking.

Here we explored the potential of proteomic profiles to inform the prediction of multi-disease and mortality risk (Fig. 1). We constructed a disease/mortality-specific proteomic risk score (ProRS) based on 1461 Olink plasma protein measurements for 45 conditions comprising infectious, blood, endocrine, mental, neurological, sensory, circulatory, respiratory, digestive, skin, musculoskeletal, and genitourinary diseases, cancers, and mortality. We extensively explored the proteomic profiles by incorporating them into Cox proportional hazard regressions, modeling the risk for individual outcomes, and comparing the predictive power of ProRS with that of established clinical predictors. Moreover, we investigated the shared proteomic profiles and the clinical utility of proteomics across multiple incident diseases and mortalities.

## Results

### Study population

This study adopted 52,006 participants with blood proteomics data currently available in the UK Biobank (UKB), and the population had a median age of 58 years (interquartile range (IQR) 50–64), of whom 53.9% were female, and mainly consisted of white ethnicity (93.7%). Median years of education were 11 (IQR 10–15), body mass index was 26.8 (IQR 24.2–29.9), systolic blood pressure was 138.0 mmHg (IQR 126.0–152.0), and 5481 (10.6%) people were current smokers (Table 1). Detailed summary statistics and notations of all 54 clinical predictors are shown in Supplementary Data 2. During a median follow-up time of 14.1 (IQR 13.4–14.8) years until March 2023, 5625 participants died (10.82%), 7654 people developed cancer (15.76%), and the most common specific diseases were hypertension ($n = 4911$, 15.96%) and anemia ($n = 4528$, 9.31%). See Supplementary Data 1 for statistics of each endpoint.

### ProRS stratifies the risk of multiple diseases and mortality

The ProRS was derived from a single-domain assay of 1461 plasma proteomic data through an established ProNNet model. The ProNNet served as a feature extractor to translate the proteomic data into a list of 45 vectorized probabilities, named ProRS, and each probability was treated as the future incident risk corresponding to 45 endpoints, covering different categories of diseases and mortalities (Fig. 1). Participants with a higher percentile of ProRS at baseline exhibited elevated observed event rates across all 14 disease categories and all-cause mortality (Fig. 2a and Supplementary Fig. 3). Age showed a significantly positive correlation with ProRS for all 45 endpoints. Except for obesity and breast cancer, the correlation coefficients between ProRS and age exceeded 0.1, and the strongest associations were found in eye problems (correlation coefficients [95% CI]: 0.78 [0.77–0.78]), cancer (0.70 [0.70–0.71]), and circulatory system disorders (0.67 [0.66–0.68]). Significant differences existed between ProRS scores in males and females across 42 endpoints, except for prostate cancer, breast cancer, and inflammatory bowel disease. Among individuals at the same percentile of ProRS, males had a notably elevated risk of cancer, circulatory system disease, and all-cause mortality when compared to females (Fig. 2a, Supplementary Data 4).

The Kaplan-Meier survival curves showed distinctive paths between the tertiles stratified by ProRS (Fig. 2b and Supplementary Fig. 4). Compared with the bottom tertile, individuals with ProRS in the top tertile had more than fivefold elevated risk of all-cause mortality (odds ratio (OR) [95% confidence interval (CI)]: 11.83 [10.11–13.55]) and blood and immune disorders (5.08 [4.66–5.50]). In contrast, the ORs were much smaller for skin disorders (1.82 [1.50–2.14]) and digestive system disorders (1.65 [1.48–1.82]). Within a detailed classification scheme, the top tertile of ProRS resulted in event rates more than fivefold higher compared with the bottom tertile across 16 out of 26 specific diseases and all four causes of mortality (Supplementary Data 5). Of note, the OR exceeded 20 in death caused by the respiratory system (OR [95% CI]: 53.80 [38.17–69.43]), dementia (32.83 [21.18–44.48]), death caused by the circulatory system (29.51 [21.68–37.34]), diabetes (23.58 [19.36–27.79]), and obesity (20.45 [15.00–25.89]).

We investigated the predictive abilities of ProRS across different time windows, for most endpoints, the ProRS achieved the highest area under the ROC curve (AUC) when forecasting outcomes happening within 5 years, suggesting a pivotal role of plasma proteomics in detecting near-term risks. For some endpoints, the over 10-year model showed the highest AUC, including eight disease categories, three specific diseases (viral infections, neurotic disorders, and sleep disorders), and death caused by the nervous system (Supplementary Data 6).

### Discriminative improvements over clinical predictors

We investigated the predictive information of ProRS and three clinical predictor sets with increasing complexity (including Age+Sex, Serum (25 serums), and PANEL (all 54 clinical predictors)) for each endpoint.

The protein-only model, the CPH model fitted with ProRS only, yielded Harrell's C-indexes greater than 0.80 in all-cause mortality, two causes of death (respiratory system and circulatory system), and seven specific diseases, including diabetes, lung cancer, prostate cancer, dementia, obesity, chronic obstructive pulmonary disease, and renal failure. In most endpoints, ProRS alone had significantly greater or comparable discriminative performance than Age+Sex, Serum, and PANEL. Furthermore, the ProRS significantly outperformed all three sets of clinical predictors in particular diseases, including five disease categories (diseases of infections, blood and immune disorders, nervous system disorders, respiratory system disorders, and genitourinary system disorders), seven specific diseases (bacterial and viral infections, leukemia, anemia, dementia, heart failure, and chronic obstructive pulmonary disease), and all-cause mortality and its four causes (Fig. 3, Supplementary Data 7). It demonstrates that ProRS generally contains more competitive predictive information than Age+Sex, Serum, and PANEL.

When incorporating ProRS into the Age+Sex or Serum models, a significant enhancement in predictive capability was observed across almost all endpoints (13 disease categories, all-cause mortality, 23 specific diseases, and four causes of death in the Age+Sex model; all endpoints in the Serum model), but the combination did not

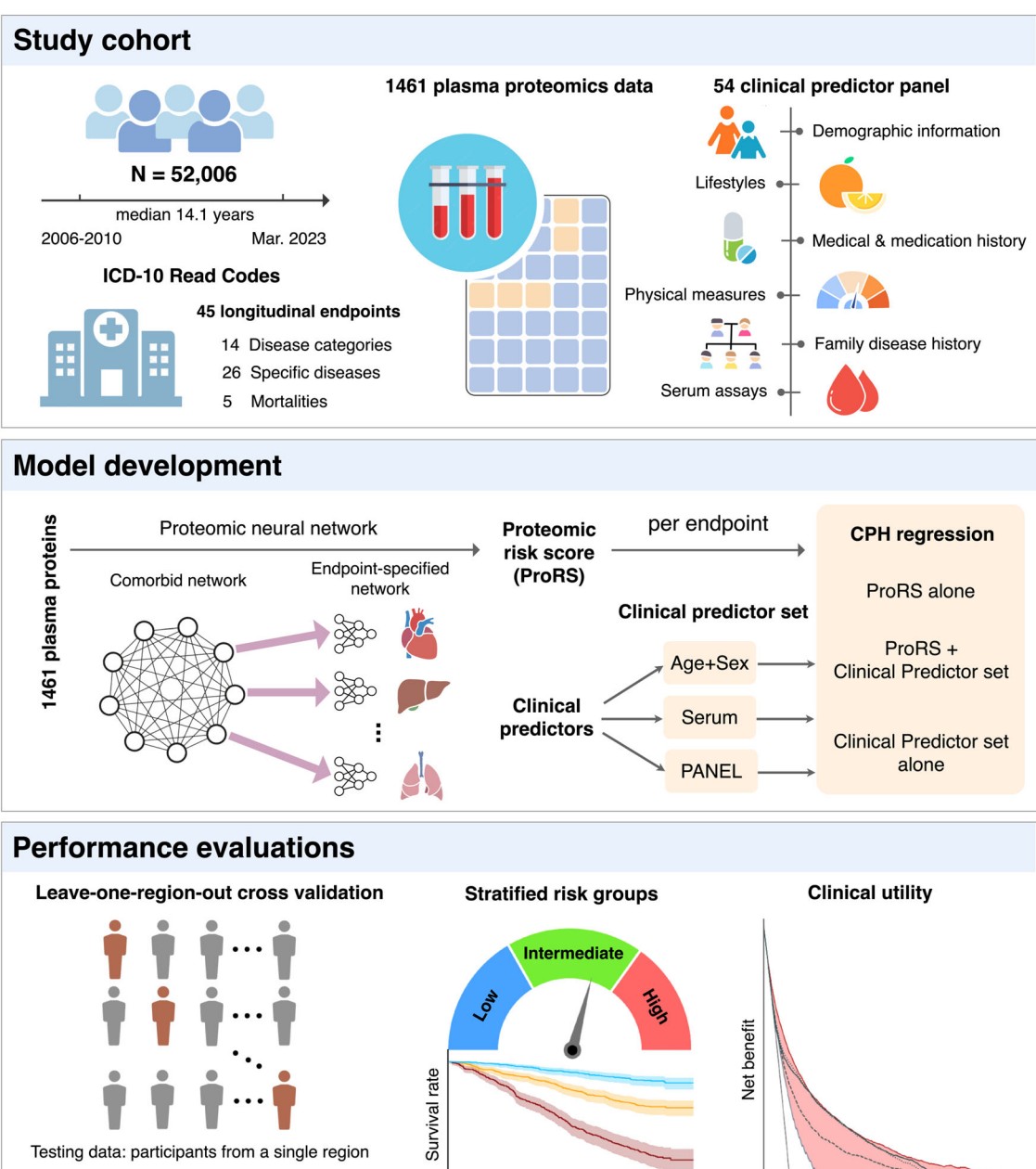

**Fig. 1 | Study overview.** First, we extracted data from 52,006 UKB participants with a median follow-up time of 14.1 years, including 45 endpoints defined by three-character ICD10 codes, 1461 plasma proteomics, and 54 clinical predictors spanning demographic, lifestyles, physical measures, medical and medication history, family disease history, and serum assays. Next, we developed a proteomic neural network to generate proteomic risk scores (ProRS) for each endpoint. Downstream survival analysis was performed using Cox proportional hazard models to explore ProRS and clinical predictor sets individually or in combinations. The model establishment and evaluations were implemented through internal leave-one-region-out cross-validation. Our investigation not only focused on evaluating models' efficacy in stratifying populations at risk but also aimed to inform their potential clinical utility. CPH model Cox proportional hazard model, ProRS proteomic risk score.

significantly exceed ProRS alone in most endpoints. Of note, the protein-only model exhibited significantly improved discrimination in predicting breast cancer, prostate cancer, leukemia, dementia, Parkinson's disease, all-cause mortality, death caused by the nervous system, death caused by the circulatory system, and death caused by the respiratory system when compared to the combination of Serum and ProRS.

Adding ProRS to PANEL significantly improved predictive information over PANEL for 11 disease categories, all-cause mortality, 20 specific diseases, and four causes of death. It's worth noting that, in more than one-third of endpoints, the combination of ProRS and PANEL produced comparable C-indexes to ProRS alone. For the remaining endpoints, combining PANEL with ProRS yielded improved prediction performance compared to models based solely on single-domain source data. However, the extent of the enhancement in predictive capabilities was limited when compared to using ProRS alone. In addition, we examined the impact of plasma proteomics on diseases in the presence of established risk factors. After adjusting PANEL in the Cox proportional hazard models, ProRS was still significantly associated with 13 disease categories, all-cause mortality, 24 specific diseases, and four causes of death (Supplementary Data 8).

**Table 1 | Participants' characteristics of UK Biobank**

| Participants characteristics | All participants (N = 52,006) | Female (N = 28,056) | Male (N = 23,950) |
|---|---|---|---|
| **Demographic** | | | |
| Age | 58 [50–64] | 58 [50–63] | 59 [50–64] |
| Ethnicity (white) | 48,517 (93.7%) | 26,204 (93.8%) | 22,313 (93.7%) |
| Townsend deprivation index | −2.1 [−3.6 to 0.8] | −2.1 [−3.6 to 0.7] | −2.1 [−3.6 to 0.9] |
| Education, years | 11 [10–15] | 11 [10–15] | 11 [10–15] |
| **Lifestyle** | | | |
| Current smoker | 5481 (10.6%) | 2492 (8.9%) | 2989 (12.5%) |
| Frequent alcohol intake | 35,615 (68.6%) | 17,248 (61.6%) | 18,367 (76.9%) |
| Regular activity | 39,548 (78.0%) | 21,094 (77.2%) | 18,454 (78.9%) |
| **Physical measures** | | | |
| Body mass index | 26.8 [24.2–29.9] | 26.2 [23.5–29.8] | 27.3 [25.0–30.0] |
| Basal metabolic rate, kj | 6360 [5489–7581] | 5552 [5184–6000] | 7657 [7037–8355] |
| Systolic blood pressure, mmHg | 138 [126–152] | 135 [122–150] | 141 [130–154] |
| **Disease and medication history** | | | |
| History of diabetes | 2948 (5.7%) | 1098 (3.9%) | 1850 (7.7%) |
| History of hypertension | 14,605 (28.1%) | 6745 (24.0%) | 7860 (32.8%) |
| Cholesterol medications | 9678 (18.9%) | 3747 (13.5%) | 5931 (25.2%) |
| **Family disease history** | | | |
| Parents' history of diabetes (single) | 7565 (16.7%) | 4293 (17.2%) | 3272 (16.1%) |
| Parents' history of diabetes (both) | 603 (1.3%) | 319 (1.3%) | 284 (1.4%) |
| Parents' history of dementia (single) | 5421 (12.0%) | 3132 (12.6%) | 2289 (11.2%) |
| Parents' history of dementia (both) | 222 (0.5%) | 121 (0.5%) | 101 (0.5%) |
| **Serum** | | | |
| HDL cholesterol, mmol/L | 1.4 [1.2–1.7] | 1.5 [1.3–1.8] | 1.2 [1.1–1.4] |
| Triglycerides, mmol/L | 1.5 [1.0–2.1] | 1.3 [1.0–1.9] | 1.7 [1.2–2.4] |
| Glucose, mmol/L | 4.9 [4.6–5.3] | 4.9 [4.6–5.3] | 5.0 [4.6–5.4] |
| HbA1c, mmol/mol | 35.3 [32.9–38.1] | 35.3 [32.9–37.9] | 35.4 [32.8–38.3] |
| IGF-1, nmol/L | 21.2 [17.4–24.7] | 20.6 [16.8–24.4] | 21.7 [18.1–25.1] |
| Alanine aminotransferase, U/L | 20.0 [15.3–27.1] | 17.5 [13.9–22.8] | 23.4 [18.2–31.4] |
| Albumin, g/L | 45.1 [43.4–46.9] | 44.9 [43.2–46.6] | 45.4 [43.7–47.2] |
| Creatinine, umol/L | 70.7 [61.4–81.4] | 63.2 [57.0–70.5] | 80.2 [72.6–88.9] |
| Leukocyte count, $10^9$ cells/L | 6.6 [5.6–7.9] | 6.6 [5.6–7.8] | 6.7 [5.6–7.9] |
| Erythrocyte count, $10^{12}$ cells/L | 4.5 [4.2–4.8] | 4.3 [4.1–4.5] | 4.7 [4.5–5.0] |
| Platelet count, $10^9$ cells/L | 248.0 [213.0–287.0] | 261.0 [225.2–300.0] | 233.2 [201.0–269.0] |

Continuous data are described as median [interquartile range], and categorical variables are presented as numbers (percentages). See Supplementary Data 2 for full clinical predictors and detailed notations.
*IGF-1* insulin-like growth factor 1, *HDL* high-density lipoprotein.

## Contributions of proteins to prediction across the spectrum of diseases

By employing the SHapley Additive exPlanations (SHAP) values, we sorted the plasma proteins based on their importance in predicting different endpoints. This enabled us to identify the most important discriminators (top 1%) associated with each condition. Some proteins play crucial roles in the prediction of various diseases. Of particular note, the GDF15, which emerged as a robust predictor across all 14 categories of disorders and all-cause mortality and even ranked first among different causes of death and seven specific diseases, namely bacterial infections, anemia, mood disorders, arrhythmias, heart failure, inflammatory bowel disease, and renal failure (Fig. 4a, Supplementary Data 9). Similarly, CDCP1, CXCL17, EDA2R, and HAVCR1 demonstrated important predictive value across more than ten disease categories. The direction of associations between these proteins and different diseases remained consistent (Fig. 4b). In contrast, NEFL, BCAN, TNFRSF10B, and CA14 exhibited importance in relatively fewer disease categories. Proteins such as NTproBNP, TSPAN1, and ACE2 were deemed important only in two disease categories.

Subsequently, we focused on cancer and dementia, two diseases receiving much attention in recent decades[30] (Fig. 5). Proteins including CXCL14, GDF15, HAVCR1, and CDCP1 were identified as predominant contributors to the risk of cancer. We confirmed that higher plasma levels of CXCL14, GDF15, HAVCR1, CDCP1, TSPAN1, LTBP2, and ACTA2 were associated with higher risk, while higher plasma levels of RET showed a protective effect. For dementia, NEFL, BCAN, GFAP, and GDF15 were the main proteins influencing disease risk. Consistent with previous findings, we observed the risk effects of CDCP1[31, 32], EDA2R[33], and HAVCR1[34]. Further, we identified ACTA2, LTBP2, and NCS1 as potential contributors to dementia risk. Comprehensive data for all investigated endpoints can be found in Supplementary Data 9.

## Model calibration and clinical utility

The ProRS showed great performance in discriminating populations at risk, but such results could not implicate whether ProRS should be used in clinical practice. To provide a statistic of immediate clinical interpretability, we further assessed the predictive models by examining calibrations and conducting decision curve analysis. Except for viral infections, models for almost all endpoints were well calibrated, where the observed risk and the predicted risk showed consistency (Fig. 6a–d, Supplementary Figs. 5 and 6). Figure 6e–h and Supplementary Figs. 7 and 8 display the net benefit curves, showing the tradeoffs of benefits and harms for clinical decisions based on different models across a range of decision thresholds.

We specifically investigated the clinical utility of ProRS in two scenarios. First, we evaluated the performance of ProRS and two sets of clinical predictors with distinct complexity (Age+Sex and PANEL) individually. Second, we combined the ProRS with these two sets of clinical predictors to uncover potential net benefits. For most diseases, the protein-only model showed greater net benefit compared with the models solely based on Age+Sex or PANEL. The addition of ProRS to Age+Sex or PANEL substantially improved the clinical utility, but the combination performed similarly to ProRS alone. Within a certain range of decision thresholds, ProRS even demonstrated a higher net benefit compared to the combination approach (most notably all-cause mortality and, to a lesser extent, dementia and heart failure). In contrast, if ProRS did not surpass these two sets of clinical predictors in discrimination, no superiority or improvement in clinical utility could be observed, as is the case in cancer (Fig. 6e).

## Discussion

Leveraging systematic information from proteomic profiling, we found that the integrative proteomic status markedly stratified the risk for 45 common conditions, including infectious, hematological, endocrine, psychiatric, neurological, sensory, circulatory, respiratory, digestive, cutaneous, musculoskeletal, and genitourinary diseases, cancers, and mortality. Notably, ProRS yielded much better or comparable predictive performance than conventional clinical predictors for almost all endpoints, and combining clinical variables with ProRS did not demonstrate substantial advantages over ProRS alone. Some proteins, such as GDF15, exhibited important discriminative values for

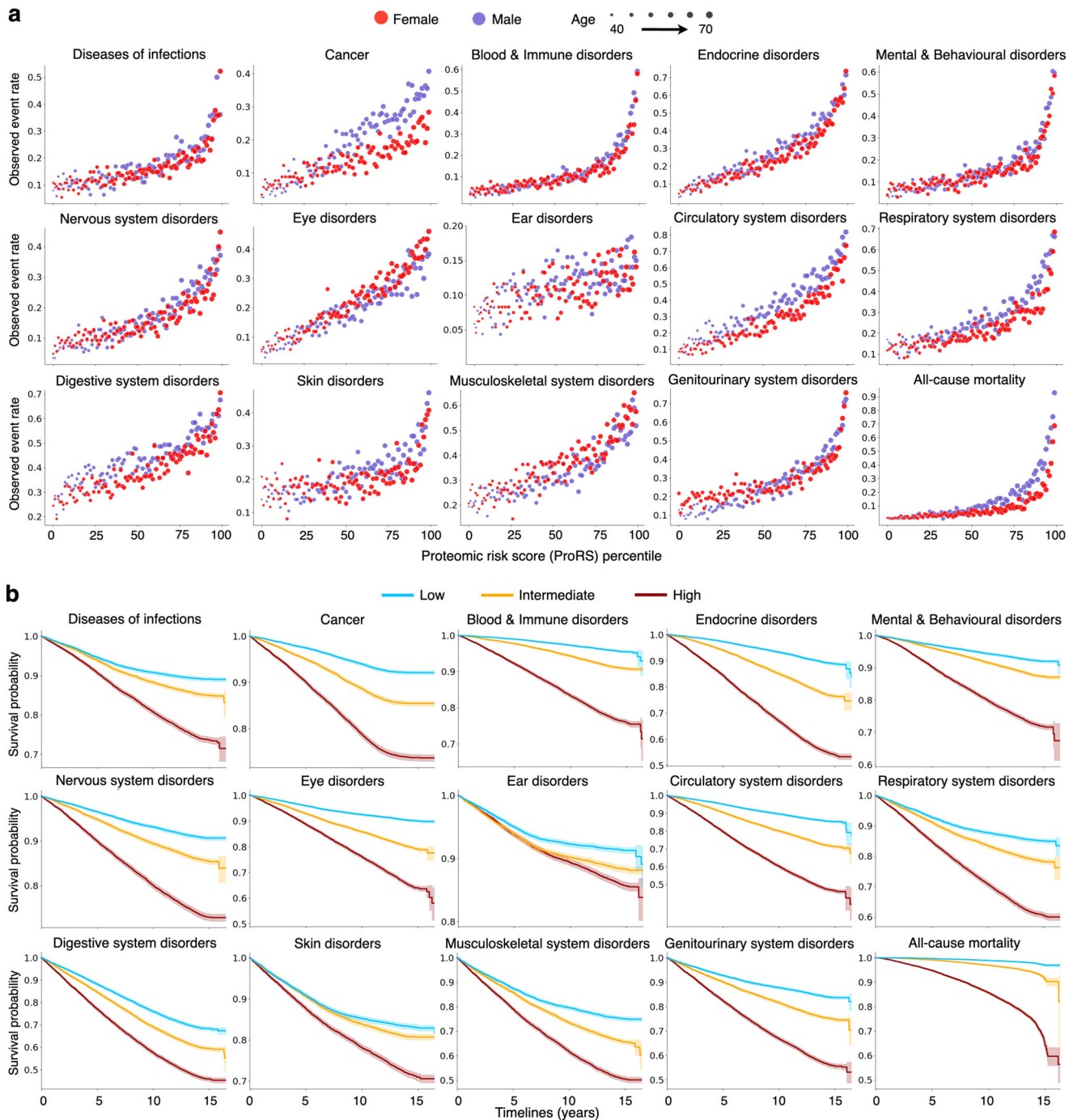

**Fig. 2 | ProRS is associated with observed event rates and stratifies survival in multiple diseases. a** Observed event frequency for 14 incident disease categories and all-cause mortality plotted against ProRS percentiles. Age is represented by varying sizes of dots, while sex is distinguished by different colors of the dots. **b** Kaplan-Meier survival plots of 14 incident disease categories and all-cause mortality according to tertiles of ProRS during follow-up (blue, bottom tertile; yellow, median tertile; brown, top tertile) with 95% exponential Greenwood confidence intervals. Source data are provided as a Source Data file. ProRS proteomic risk score.

various diseases. In addition, and importantly, these discriminatory improvements could be translated into clinical utility for a wide range of potential decisions.

A single plasma protein is unlikely to predict the risk of multiple disease events simultaneously. Overcoming the hurdles of proteomic techniques, the risk score we constructed, based on 1461 proteins, stratified risk trajectories well for all endpoints studied. We modeled the score on the process preceding the onset of the disease rather than on the outcome of clinical treatment, which is essential for the early identification of individuals in need of more intensive therapeutic

interventions and for significantly advancing the window for interventional treatment. Patients' age is an important component of risk stratification, as older patients are at higher risk of multiple morbidities and death[35]. Our score does not rely on people's age, but a higher ProRS typically indicates an older age and a higher risk of adverse disease outcomes. Notably, among people at the same percentile of ProRS, men had a higher risk of cancer, circulatory system disease, and all-cause mortality than women, consistent with the view that sex and disease are interrelated[36]. The male predisposition to cancer may be affected by the genetic programming of male cells, the post-pubertal

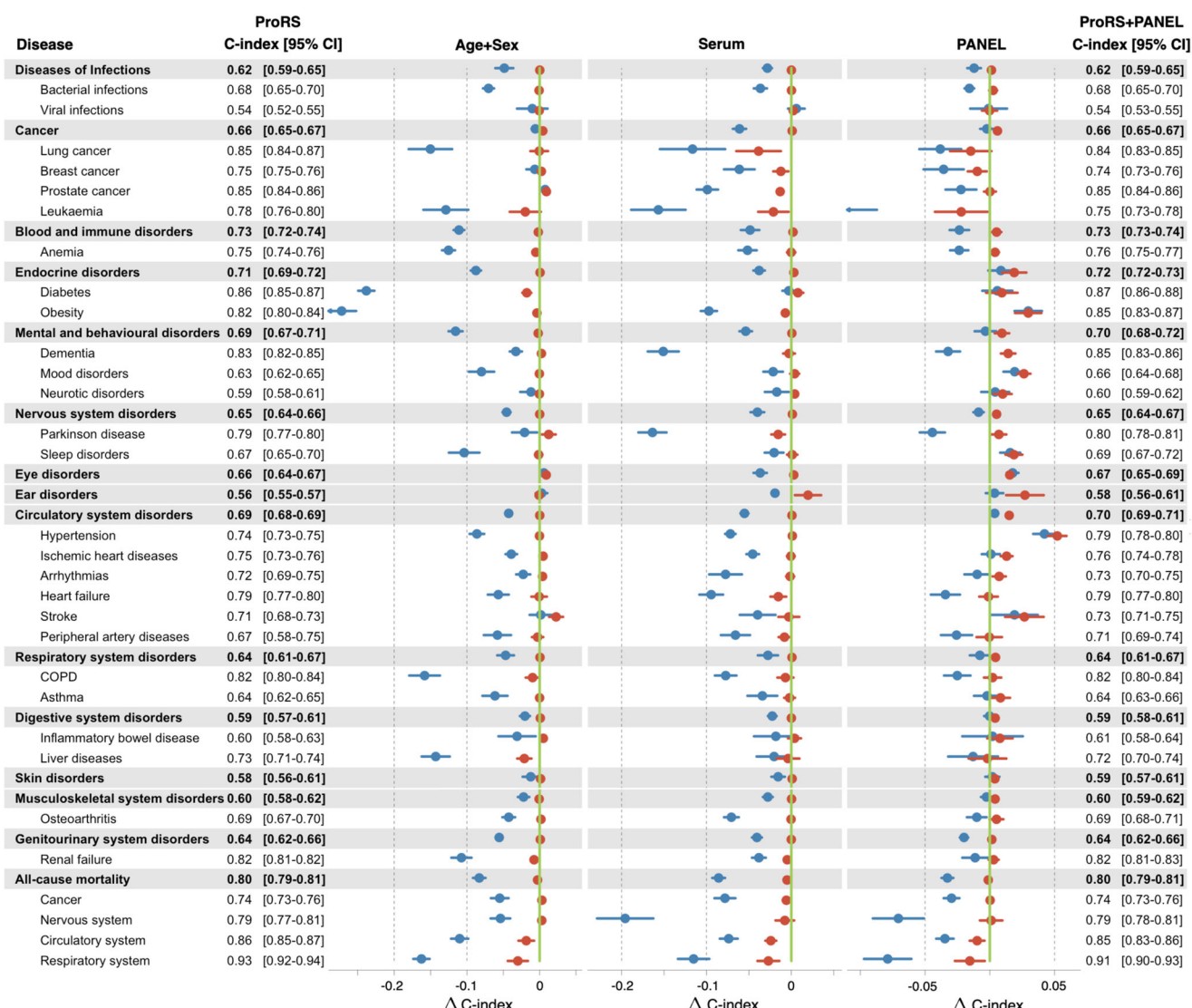

**Fig. 3 | Predictive value of the ProRS in comparison with clinical predictor sets for multiple diseases.** Forrest plot showing the differences in discriminative performance of ProRS (vertical green line), clinical predictor sets only (blue), and the combination of ProRS and clinical predictor sets (red). Models' performance is presented using Harrell's C-index, where dots represent the means and horizontal bars represent the 95% confidence intervals derived from cross-validation. Number of available participants for each endpoint is listed in Supplementary Data 1. The clinical predictors included three sets with increasing complexity: Age+Sex, Serum (25 serum measures), and PANEL (all 54 clinical predictors). Source data are provided as a Source Data file. COPD chronic obstructive pulmonary disease.

sex hormones, and the interactions with sex-specific behaviors (e.g., smoking, lifestyle, perceived stress and distress, and dietary habits)[36]. Lipid markers and depression may influence the risk of circulatory disease in men[37]. Given the current lack of appreciation for personalized disparities and the demands of precision medicine, endeavors to incorporate age and sex differences into guidelines for risk stratification and disease prevention are urgently required.

In line with previous reports demonstrating the high predictive value of proteomic profiling for multiple health issues, such as cardiovascular events[5,14,38], obesity[15], dementia[39], and cancer[40,41], our study extends these findings to diseases of all human systems. To our knowledge, this is the first time that the predictive power of blood proteomics for a broad range of disease outcomes has been extensively revealed, not only by comparison with a diverse panel of clinical variables but also by investigating the additional predictive ability of proteomics beyond clinical predictors. Importantly, we found that ProRS has much better or comparable predictive performance than established clinical indicators for almost all investigated endpoints. Besides, ProRS alone could achieve desirable predictions for most

endpoints, with predictive power comparable or close to that of the model combining ProRS and clinical indicators. ProRS even outperformed the recently proposed metabolomic profile, where the predictive power of metabolomic states alone was not superior to that of the combination of age and sex[4]. Our data strongly emphasize the predictive value of blood proteomics as a single-source, individualized health check tool, alleviating the hassle of multidimensional data collection required for previously proposed risk scores[38]. Proteomics analysis may therefore have great promise as an alternative to complex laboratory tests or clinical evaluations to improve the assessment of risk for multiple diseases simultaneously.

Plasma biomarkers reflecting human health status are central to clinical decision-making. The biomarkers we routinely use tend to target specific diseases or disease spectrums. Since we found that certain proteins were associated with different diseases in a highly concordant direction, interventions on identified shared pathways are likely to deliver benefits in a consistent manner without increasing the risk of developing other diseases[42]. In this scenario, GDF15 may serve as an attractive target for disease intervention given its robust predictive

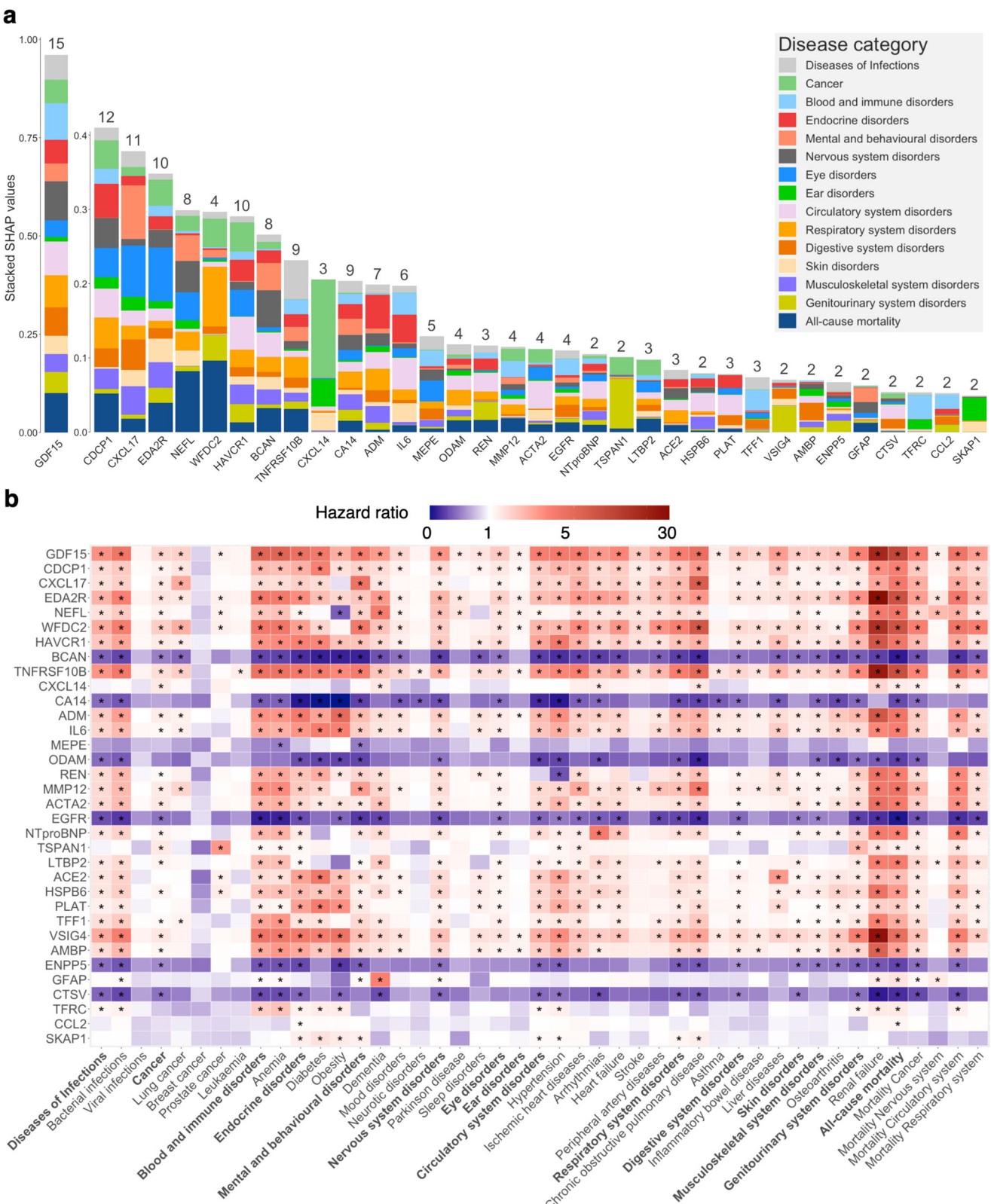

**Fig. 4 | Proteins with the most important discriminative value (top 1%) and their associations with each endpoint. a** Stacked bar chart of standardized SHAP values from ProNNet across 14 disease categories and all-cause mortality, numbers on top of the bars indicate how many disease categories in which this protein showed the predominant discriminatory significance. We highlighted 34 proteins that exhibited the most important discriminatory value in two or more disease categories. **b** Associations between proteins are shown in (**a**) and 45 endpoints. The color of cells indicates the effect size (HR) between each protein and incident endpoint. HR was derived based on normalized proteins fitted using Cox proportional hazard models adjusted with age and sex. Asterisks in cells represent significant associations after correction for multiple comparison testing (*p*-value < 6.84 × 10⁻⁶ = 0.01/1461, *p*-value was derived corresponding to a two-sided test). Source data are provided as a Source Data file. HR hazard ratio, SHAP SHapley Additive exPlanations.

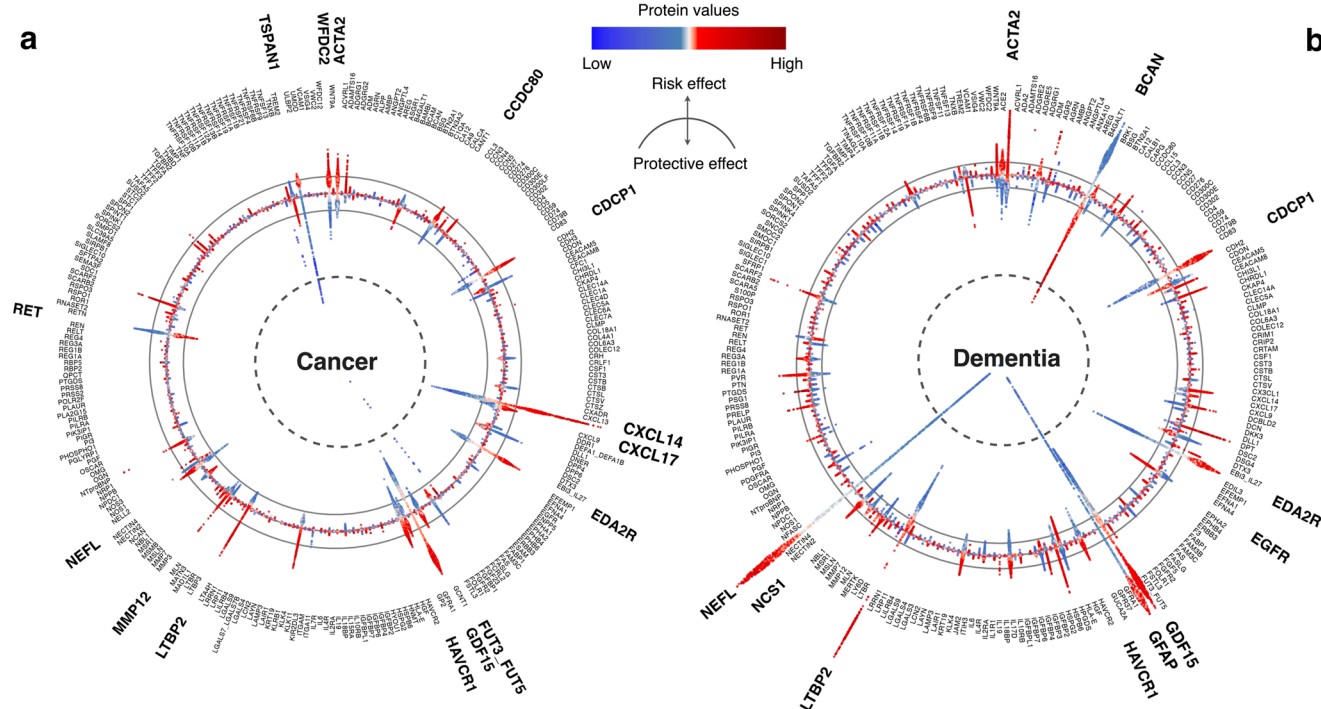

**Fig. 5 | Plasma proteins attributions for cancer and dementia.** The circular SHAP plots of cancer (**a**) and dementia (**b**) show the extent to which each protein would affect the proteomic neural network model. Proteins shown in the figure were pre-selected through age- and sex-adjusted CPH models under Bonferroni correction ($p$-value < 6.84 × 10$^{-6}$ = 0.01/1461, $p$-value was derived corresponding to a two-sided test). The important proteins ranking in the top 1%, as determined by SHAP values, are bolded to emphasize their significance in disease prediction. The width of the range of the bars indicates the extent of proteins' contribution to disease prediction, with wider bars reflecting a greater contribution. The color of the bars represents the magnitude of plasma proteins, ranging from low (blue) to high (red), as depicted in the color bar above the middle. Deviations toward the center and periphery signify protective and risk contributions, respectively. Noted, four of the top 15 proteins selected by SHAP were not shown in dementia as they did not surpass multiple testing, probably due to their nonlinear effect or high correlations with adjusted covariates. Source data are provided as a Source Data file. CPH model Cox proportional hazard model, SHAP SHapley Additive exPlanations.

power for all disease categories and all-cause mortality. As mentioned previously, elevated GDF15 represented a predictor for the future development of cardiovascular disease[43], diabetes[44], chronic kidney disease[45], adverse respiratory outcomes[46,47], dementia[48], cancer[49], and all-cause, cardiovascular, and non-cardiovascular mortality[50,51], and was invariably linked to poor prognosis[52]. Our findings further expanded the detrimental effects of elevated GDF15 on all body systems. This is plausible in that the expression of GDF15 could be induced in response to cellular stress and mitochondrial dysfunction in order to maintain cellular and tissue homeostasis[53], and we hypothesize that this will play a role in the pathogenesis of various disease processes. The predictive value of CDCP1[54], CXCL17[55], and EDA2R[56] for diseases has previously focused on cancer, and HAVCR1 has been shown to be involved in immunity and renal regeneration and is aberrantly expressed in a variety of tumor types[57]. We provide preliminary evidence for the predictive value of these proteins in multiple diseases, and future relevant studies are needed to verify our observations.

With growing evidence that dementia prevention might be attainable by modifying risk factors[58], our findings are complementary to this work by spotting populations that would benefit most from preventive interventions. We confirmed the associations of NEFL[59], BCAN[60], GFAP[61], and GDF15[48] with dementia and suggested that the abnormal levels of these proteins may imply a high risk of developing dementia. For cancer, CXCL14 showed the greatest predictive importance. The levels of CXCL14 expression were closely linked to some clinicopathologic factors, including tumor-node-metastasis stage, tissue differentiation, and tumor size, which have heretofore been noted as possible predictors for the early recurrence and death of cancer[62].

Good discrimination and calibration across various prediction horizons for each outcome are paramount to the clinical utility of risk models[63]. As one of the largest and most comprehensive population cohorts in the world, UKB enables us to evaluate clinical usefulness with high precision and powerful efficacy[64]. In the current study, the proteomics-based scores not only exhibited favorable discriminative performance for an array of diseases but all models were well calibrated to guarantee the reproducibility and reliability of the results. We also proved that the observed discriminative gains could be translated to practical utility gains. The findings provide strong evidence for the promotion of proteomics in clinical practice, which could help improve risk assessment for numerous diseases from a holistic perspective of human health and further contribute to the implementation of targeted disease prevention strategies and tailored treatments.

This study has numerous advantages, including the application of high-throughput proteomic technologies, the large community-based prospective cohort sample, the long follow-up period, and the comprehensive assessment of disease outcomes. Several limitations also merit discussion. First, some proteins that were not included in the Olink panels but could predict multi-disease outcomes may have been omitted. Yet, the purpose of this study was to assess the feasibility of the proteomics-based risk score for clinical use, not to discover novel proteins. However, we cannot rule out the possibility that the predictive value of a larger protein panel may be superior. Second, the incidence of disease events is probably lower than that in other reported cohorts given that the UKB population tends to be healthier and less deprived[64]. Nevertheless, the large cohort size and the heterogeneity of exposures allow for valid scientific inferences about proteomic-disease relationships that are widely generalizable[64]. Third, the majority of UKB participants are white and European. Whilst we

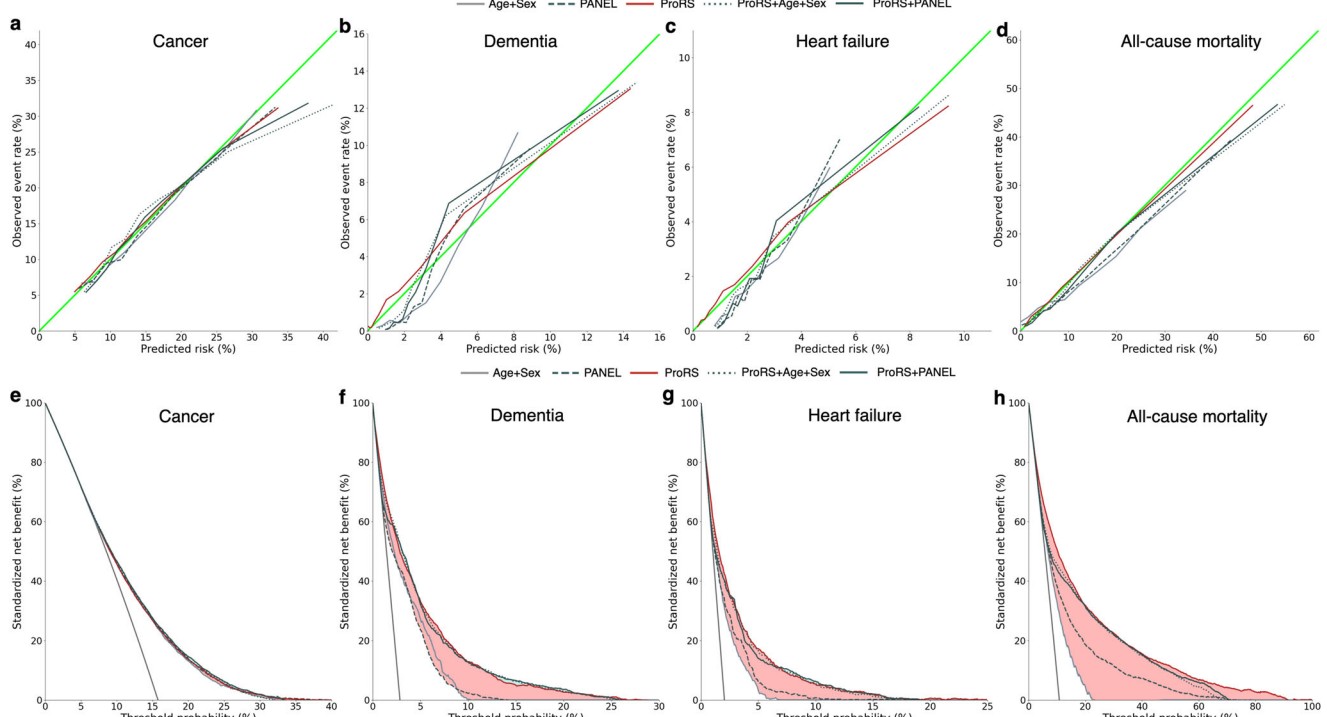

**Fig. 6 | Model calibration and predictive value of ProRS translate to potential clinical utility. a–d** Calibration curves for CPH models, including three models based on ProRS and two clinical predictor sets (Age+Sex and PANEL) individually, and two models based on their combinations (Age+Sex+ProRS, PANEL+ProRS) for the endpoints cancer (**a**), dementia (**b**), heart failure (**c**), and all-cause mortality (**d**). **e–h** Endpoint-specific net benefit curves standardized by endpoint prevalence, where vertical solid gray lines indicate 'treat all'; cancer (**e**), dementia (**f**), heart

failure (**g**), and all-cause mortality (**h**). The standardized net benefit of ProRS is compared with Age+Sex or PANEL. We also compare the combination (Age+Sex +ProRS, PANEL+ProRS) with ProRS. The red color-filled area indicates the gains in benefit achieved by ProRS compared with clinical predictors. The green color-filled area indicates the added benefit of the combination compared with ProRS. Source data are provided as a Source Data file. CPH model Cox proportional hazard model, ProRS proteomic risk score.

conducted internal cross-validations and all models were well calibrated, our findings warrant validation in a large, unselected, ethnically diverse primary care population using an independent external dataset before application in routine care.

Blood proteomics demonstrated tremendous advantages in improving risk stratification and possessed desirable predictive performance for a wide range of diseases and mortalities, even outperforming established clinical predictors. These discriminatory values could also be largely translated into practical clinical utility. Taken together, our work underscores the critical potential of proteomic profiling as a single-domain assay to inform the risk of multiple diseases and mortalities simultaneously.

## Methods

### UK Biobank study cohort

We performed a retrospective study by extracting data from the UK Biobank (UKB) cohort, a global biomedical database of half a million UK participants aged 40–69 years at baseline. Participants were enrolled from 2006 to 2010 in 22 recruitment centers across the UK. A randomized subset of UKB participants was conducted proteomic profiling on blood plasma samples collected from UKB participants' baseline recruitment. We excluded those over 30% of missingness in proteomics data and finally included 52,006 participants who had a median follow-up of 14.1 years until March 2023. The study was conducted following the Declaration of Helsinki. All participants provided written consent, and approval was given by the North West Multicentre Research Ethics Committee (MREC, https://www.ukbiobank.ac.uk/learn-more-about-uk-biobank/about-us/ethics). This research has been conducted using the UK Biobank Resource under approved application number 19542.

This study adopted 45 endpoints, including 14 disease categories, 26 specific diseases, all-cause mortality and 4 cause-specific mortalities. Endpoints were ascertained and classified according to the ICD 10 codes (Supplementary Data 1), extracted from first occurrences data (UKB category 2401-2417) that mapped from data sources of primary care (category 3000), hospital inpatient (category 2000), self-reported medical conditions (UKB field 20002) and death register records (field 40001 and 40002). Participants' follow-up started from the date of their first visits to the UKB assessment centers (baseline time), the same time that blood samples and other clinical information were collected, and participants' follow-up was censored upon the earliest date of disease diagnosis, death, or the last available date from the hospital or general practitioner, whichever occurred first (censored time). Exclusion criteria for each endpoint were defined as any diagnosis sourced from self-reported clinical records or any incidents indexed before the baseline of the respective disease category that the endpoint belongs to.

### Blood proteomics and clinical predictors

UK Biobank Pharma Proteomics Project consortium generated blood-based proteomic data. Blood samples were collected in EDTA (9 mL) vacutainers, and fractioned to 850 μL aliquots of EDTA plasma, buffy coat, and red cells. The plasma samples were stored in a −80 °C freezer before being shipped on dry ice to Olink Analysis Service in Sweden. Proximity Extension Assay, in combination with Next-Generation Sequencing[65], was utilized to parallelly measure 1463 unique proteins from April 2021 through January 2022. Following stringent quality control (see details in biobank.ndph.ox.ac.uk/ukb/ukb/docs/PPP_Phase_1_QC_dataset_companion_doc.pdf), proteins were measured across four panels containing cardiometabolic, inflammation, neurology, and

oncology proteins. Details on sample selection, in addition to processing and quality control information for the Olink assay, are provided in previous publications[66,67]. We finally included 1461 unique proteins in our study after the exclusion of those with missingness over 50%. Supplementary Data 10 lists all proteins used in this study.

To investigate the predictive ability of proteomics incorporated with other accessible measures, the study adopted a wide range of clinical variables ($n = 54$) collected at baseline, including demographic information ($n = 5$), lifestyle factors ($n = 6$), physical measurements ($n = 7$), disease and medication history ($n = 7$), family disease history ($n = 4$) and serum biochemistry data ($n = 25$). Specifically, we delivered three sets of variables, Age+Sex, Serum (25 serums) and PANEL (all 54 variables). See Supplementary Data 2 for all clinical predictors adopted in this study.

## ProNNet and ProRS

To exploit the potential of plasma proteomics as a single-domain assay to simultaneously predict multi-endpoints onset, we developed a multilayer perceptron neural network and named it ProNNet. The ProNNet model comprises two modules: a comorbid network to roughly evaluate the overall health conditions and an endpoint-specified network to customize the risk prediction task to each endpoint (Supplementary Fig. 2). The comorbid network contains two identical branches with output targets as the total number of the 14 disease categories indexed before (left branch) and indexed after (right branch) the baseline assessments. In other words, the comorbid network was trained to learn how many disease categories have been indexed in the past and will be indexed in the future. Features derived from the comorbid network learnt information of estimating the overall level of an individual's health conditions; thus, such latent features could donate potentially contribution to the endpoint-specific prediction task. The input of the comorbid network was 1461 vectorized plasma proteins and it passed into two identical branches that contained four fully connected dense layers that each connected with ReLu activations[68] of 512, 256, 128 and 64 nodes, respectively. As the number of disease category was a continuous variable, linear activation was adopted as the final activation function. The performance of the comorbid network yielded mean absolute errors (MAEs) of $1.84 \pm 0.32$ and $1.41 \pm 0.10$ for the number of comorbidities before and after the baseline, respectively, and fitted results were shown in Supplementary Fig. 9. The comorbid network was pre-trained, and its frozen weights of the last layers were passed into the endpoint-specified network, which was designed with an encoding block like that of the comorbid network branch, and weights were concatenated in an intermediate dense layer before passing to the endpoint, ReLu activations were employed to connect the dense layers and Sigmoid activation was used before the final output layer. The loss function for the comorbid network was the summation of mean squared errors with equal weights to both branches and log loss was adopted for the endpoint-specified network. Both networks were trained with Adam optimizer[69] with a learning rate of 1e-05, batch size of 128, and epochs of 1000. To avoid overfitting, the model training process early stopped upon the epoch that no decremental of validation loss was observed after the next 25 iterations. We optimized the above hypermeters with grid search from optimizer space of SGD, RMSprop and Adam with learning rates of $10^{-1}$ to $10^{-6}$ with step of timing 0.1. ProNNet was developed using Keras (v2.7.0) under Python.

## Survival analysis of proteomic risk score and clinical predictors

We divided the participants into tertile groups based on stratified ProRS and drew Kaplan-Meier plots for each endpoint to visualize their survival curves. We also fitted Cox proportional hazard (CPH) models with different sets of clinical predictors. Specifically, we initially developed models only learned from ProRS; next, we fitted models using clinical predictors sets of Age+Sex, Serum, and PANEL; lastly, we

combined ProRS with each of the above clinical predictors set to explore its additive predictive values. Statistical significance test between paired C-index was implemented through a one-shot non-parametric approach in consideration between metrics calculated on the same sample, and this was implemented through R package CompareC (v1.3.2)[70]. CPH models were implemented with CoxPHFitter from the lifelines package (v0.27.4) under Python with a penalizer of 0.01 to facilitate model convergence.

## Proteomic attribution estimates

Shapley Additive exPlanations (SHAP)[71], a game theoretic approach, was employed to explore the proteomic attribution for each endpoint within the ProNNet model. The SHAP framework enables to identify the predictors' importance and indicates their predictive effects, either positive or negative, along with the variations of the variables' magnitude. SHAP values derived in our analysis were post-processed by dividing the summation over all 1461 proteins to allow aggregation over each fold of the testing set. We reported the top 1% proteins (15/1461 ≈ 1%) ranked based on SHAP values. SHAP values were calculated based on DeepExplainer in the package shap (v0.41.0) under Python. In addition, we also investigate the associations of proteins by using the CPH model with adjusted covariates of participants' age and sex. Bonferroni corrections were applied across all association hypothesis tests by using a family-wise error rate (FWER) of $6.84 \times 10^{-6}$ (0.01/1461) as the threshold of statistical significance under two-sided tests.

## Statistical analysis

The CPH models fitted with ProRS and clinical predictors were assessed with Harrell's C-index, which varies between 0.5 for a non-informative model and 1 for a perfectly discriminating one. Calibration plots were employed to visually depict the agreement between predicted risks versus observed event rates. Net benefit curves were drawn to observe the additive predictive value of the ProRS to different sets of clinical predictors.

We investigated the predictive value of ProRS under different incident time windows of within 5 years, within 10 years and beyond 10 years by treating incident events as binary indicators. Specifically, for evaluations within 5- and 10-year time windows, any events incident beyond 10 years after baseline were treated as no incidence. To evaluate performance on incidents beyond 10 years, participants with events indexed within 10 years were removed. We reported predictive metrics of the area under the Receiver Operating Characteristic (ROC) curve (AUC), the area under the precision-recall curve (APR) and other metrics of accuracy, sensitivity, and specificity, which were derived based on the cut-off upon achievement of the largest Youden index.

Both the ProNNet model and the following survival analysis of the CPH model were developed and evaluated using internal leave-one-region-out cross-validation. The study cohort was split based on the geographical locations of a total of 22 assessment centers, and they were merged into ten regions in the UK as our data partition criteria. Participants' statistics and demographical information summaries were reported in Supplementary Data 3. A detailed cross-validation pipeline was shown in Supplementary Fig. 1 that nine folds of data were utilized for model training and the remaining one for testing, and such a scheme was iteratively repeated until all folds had been used as both training and testing sets. Specifically, hyperparameter tuning was performed during the ProNNet training, and it was conducted using randomly partitioned fivefold cross-validation within the training set itself. Once the hyperparameters were determined, the ProNNet was then retrained based on all individuals in the training set. Next, ProRS in both training and testing sets were derived through the established ProNNet. Given pre-calculated ProRS, the following CPH model can then be established and evaluated in a cross-validation manner. It should be noted that testing sets were kept intact and solely used for model evaluations. Predictive metrics were reported with means and

95% confidence intervals calculated under ten iterative folds of testing data.

During the development of ProNNet models and CPH models, continuous variables were standardized, and categorical ones were one-hot encoded. For continuous data, missingness was imputed through the K-nearest neighbors algorithm[72], which works by identifying the nearest 50 individuals defined using Euclidean distances and imputing with their medians. Categorical data were imputed with mode. Notably, missingness was imputed for proteomic data and clinical data independently, and the procedure was implemented under each fold of data partitioned by cross-validation. All modeling and statistical analysis were implemented under Python (v3.9.16). Data imputation and statistical analysis employed the package of scikit-learn (v1.2.2).

### Reporting summary
Further information on research design is available in the Nature Portfolio Reporting Summary linked to this article.

## Data availability
The data used in the present study are available from UKB with restrictions applied. Data were used under license and are thus not publicly available. Access to the UKB data can be requested through a standard protocol (https://www.ukbiobank.ac.uk/register-apply/). Data used in this study are available in the UK Biobank under application number 19542. All data supporting the findings described in this manuscript are available in the article and in the Supplementary Information and from the corresponding author upon request. Source data are provided with this paper.

## Code availability
All software used in this study is publicly available. The code used in this study can be accessed at https://github.com/jasonHKU0907/FutureHealthProteomicPrediction.

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

## Acknowledgements

We want to thank all the participants and researchers from the UK Biobank. J.-T.Y. was funded by grants from the Science and Technology Innovation 2030 Major Projects (2022ZD0211600), National Natural Science Foundation of China (82071201), Research Start-up Fund of Huashan Hospital (2022QD002), and Excellence 2025 Talent Cultivation Program at Fudan University (3030277001). W.C. was funded by the National Natural Science Foundation of China (82071997) and the Shanghai Rising-Star Program (21QA1408700). J.-F.F. was funded by the National Key R&D Program of China (2018YFC1312904, 2019YFA0709502), Shanghai Municipal Science and Technology Major Project (2018SHZDZX01), and the 111 Project (No. B18015). The funders had no role in study design, data collection and analysis, the decision to publish or the preparation of the manuscript. Further, we would like to thank the support from the Shanghai Center for Brain Science and Brain-Inspired Technology, ZHANGJIANG LAB, Tianqiao and Chrissy Chen Institute, and the State Key Laboratory of Neurobiology and Frontiers Center for Brain Science of the Ministry of Education, Fudan University.

## Author contributions

W.C., J.-T.Y. and J.-F.F. conceived, designed, and supervised the project. J.Y. implemented model development and statistical analyses. J.Y., Y.G. and Y.Z. drafted the manuscript and accessed and verified the underlying data reported in the manuscript. J.-J.K. and L.-B.W. supported the study and contributed to the discussion of the results. All authors had

full access to all the study data and accepted the responsibility to submit it for publication.

## Competing interests

The authors declare no competing interests.
