## [Peer Review File · Nature Communications]

Reviewers' Comments:

Reviewer #1:

Remarks to the Author:

You et al performed Olinked proteomics analysis on over 52,000 samples from the UK biobank. They found that the proteomics profiles are powerful to discriminate various diseases or conditions, and to predict possible outcomes. They also found proteins, such as GDF15, as a common indicator of diseases. The manuscript is well-written. It provides enough detail about how the study was conducted, which seems very vigorous.

Overall, this reviewer thinks that the manuscript is an important contribution due to the scope of the work and their findings.

The authors have also pointed out the important weaknesses of the work, which is appreciated by this reviewer.

I only have one minor comment:

-Figure 1 is not cited or explained in the text.

This should be addressed.

Reviewer #2:

Remarks to the Author:

This paper studies 52k patients from the UK Biobank with proteomics measurements and multiple disease outcomes. Survival modeling is developed using a representation learning neural network as a feature extractor. The results show that for most diseases, survival analysis using proteomics features (in combination with traditional panels) outcompetes traditional features.

The paper is easy to follow and the analysis appears thorough and in general convincing. There are a few unclear points:

Major comments:

1. It is unclear how the feature(s) (=ProRS?) is calculated from the model output.
2. Please give more details on the cross validation procedure. Is the full neural network representation learning + survival modeling pipeline carried out leave one region out or has some of the steps been carried for all data? Please also give the exact number of patients in each fold.

Minor comments/questions:

3. Why is the network called a residual network? No residual connections are used.
4. Github link does not work.
5. "It outputs of two outcomes, followed by linear activations, are the overall number of the 14 disease categories indexed before and after the baseline assessments." This sentence is unclear. Both the modeling and data aspects should be explained better. What does before and after baseline mean here? Does this mean that each patient can appear with different covariates for different outcomes?
6. "Thus, the comorbid network is capable of 366 overwhelmingly estimating the health conditions of each individual." Meaning unclear. Can this network predict the disease outcome with high accuracy on the test data?
7. "...with early stopping criteria that no validation loss after 25 epochs" Missing word here?

Reviewer #3:

Remarks to the Author:

The authors present a new index of disease risk called ProRS, based on proteomic measures, and show that it outperforms some commonly used predictors of various diseases in a subset of the UK BioBank data set. While I found the research presented in the paper impressive, I have a number of comments that I hope may improve the work. These are listed in roughly descending order of importance.

1. The authors point out that many existing predictive models in proteomics have been cross sectional, which clearly is not ideal for prediction. However, it is not clear to me how they are addressing this issue. Does the UK biobank have longitudinal measurements? If so, the authors should clarify when the proteomics measurements were taken relative to the baseline for their time-to-event analyses.
2. I would like to see more detail about the model testing process, maybe added to Figure 1 in the paper. It is not clear what segment of the data was held out for testing the predictive models, so a diagram would be helpful. Alternatively if the only validation was cross-validation, then the authors should be explicit about that.
3. I am concerned about the claimed Bonferroni correction applied to the Cox model results. Did the authors in fact adjust the p-values or did they just use a lower threshold of 0.01 on the raw p-values? If the former, then they should clarify what denominator they used and how they define a "family" of tests when controlling the type-1 error rate; if the latter, then this is surely not adequate to handle the multiplicity of tests performed.
4. The statistical methods used to compare the performance of different predictive models are unclear. For example, line 139 states that ProRS had "significantly greater" performance. This is not a trivial problem as the metrics used are not simple sums of IID random variables, and comparing them additionally requires taking into account the correlation between metrics calculated on the same sample. The authors should provide more details about how this was accomplished.
5. The authors describe using area under the ROC curve (lines 410-411) but they apparently distinguish this from the c-index. My understanding was that the c-index (or c-statistic) was just another name for AUC. Do the authors mean something else when they use c-index?

Responses to the reviewers of Nature Communications paper NCOMMS-23-26906-T.

The authors sincerely appreciate the critical reviews of the paper, and for the helpful way in which the reviewing editors put together a constructive list of suggestions for the revision of the paper. We have now revised the paper to carefully address all the points raised, and trust therefore that the paper can now be recommended for publication in Nature Communications. Appended to this letter is our point-to-point response to the comments raised by the reviewers. The comments are reproduced, and our responses below are preceded by - -, and the changes made to the paper are indicated below within "...", and in the revised paper in red font.

Detailed response to reviewers

Response to Reviewer #1:

You et al performed Olinked proteomics analysis on over 52,000 samples from the UK biobank. They found that the proteomics profiles are powerful to discriminate various diseases or conditions, and to predict possible outcomes. They also found proteins, such as GDF15, as a common indicator of diseases. The manuscript is well-written. It provides enough detail about how the study was conducted, which seems very vigorous.

Overall, this reviewer thinks that the manuscript is an important contribution due to the scope of the work and their findings.

The authors have also pointed out the important weaknesses of the work, which is appreciated by this reviewer.

I only have one minor comment:

-Figure 1 is not cited or explained in the text.

This should be addressed.

Response: Thanks for pointing it out, we added it in the manuscript in both Introduction section (line 86), and Results section (line 112).

Reviewer #2 (Remarks to the Author):

This paper studies 52k patients from the UK Biobank with proteomics measurements and multiple disease outcomes. Survival modeling is developed using a representation learning neural network as a feature extractor. The results show that for most diseases, survival analysis using proteomics features (in combination with traditional panels) outcompetes traditional features.

The paper is easy to follow and the analysis appears thorough and in general convincing. There are a few unclear points:

Major comments:

1. It is unclear how the feature(s) (=ProRS?) is calculated from the model output.

Response: Thanks for this critical comments and we did realize a lack of characterization for this in the manuscript; thus, a description was added in the Method section (Proteomic Neural Network and Proteomic Risk Score) in lines 108-112, which reads as:

“The ProRS was derived from a single-domain assay of 1,461 plasma proteomic data through an established ProNNet model. The ProNNet served as a feature extractor to translate the proteomic data into a list of 45 vectorized probabilities, named ProRS, and each probability was treated as the future

incident risk corresponding to 45 endpoints, covering different categories of diseases and mortalities (Fig. 1).”

2. Please give more details on the cross validation procedure. Is the full neural network representation learning + survival modelling pipeline carried out leave one region out or has some of the steps been carried for all data? Please also give the exact number of patients in each fold.

Response: The authors appreciate this comments, both the Proteomic Neural Network and following survival analysis were performed and evaluated based on leave-one-region-out cross validation. To clarify the procedures step-by-step, we drew a flowchart and put it as *Supplementary Figure 1*, and we attached below for your reference. The number of participants and basic summaries of age and gender are provided in *Supplementary Table 3*. In addition, following your suggestions, we enriched descriptions with more details on cross validation procedure in the Method section (Statistical analysis) in lines 443-458 as follows:

“Both the ProNNet model and the following survival analysis of CPH model were developed and evaluated using internal leave-one-region-out cross-validation. The study cohort was split based on the geographical locations of total 22 assessment centers, and they were merged into ten regions in the UK as our data partition criteria. Participants’ statistics and demographical information summaries were reported in *Supplementary Table 3*. A detailed cross-validation pipeline was shown in *Supplementary Fig. 1* that nine folds of data were utilized for model training and the remaining one as testing, and such a scheme was iteratively repeated until all folds had been used as both training and testing sets. Specifically, hyperparameter tuning was performed during the ProNNet training, and it was conducted using randomly partitioned five-fold cross-validation within the training set itself. Once the hyperparameters was determined, the ProNNet was then retrained based on all individuals in the training set. Next, ProRS in both training and testing sets were derived through the established ProNNet. Given pre-calculated ProRS, the following CPH model can then be established and evaluated in a cross-validation manner. It should be noted that testing sets were kept intact and solely used for model evaluations. Predictive metrics were reported with means and 95% confidence intervals calculated under ten iterative folds of testing data.”

Supplementary Fig. 1: Flowchart of model development and leave-one-region-out cross-validation pipeline.

Regions were partitioned based on geographical locations of participants ‘assessment centers. A demo partition in the figure leveraging region 1 to region 9 as training set and the remaining region 10 as testing set, this partition iterative repeated until all folds of data have been used as both training and testing sets.

- a. Model development based on temporal training data.
- b. Model evaluation based on temporal testing data.

Minor comments/questions:

3. Why is the network called a residual network? No residual connections are used.

Response: Thanks for pointing it out, we previously thought both the comorbid network and endpoint-specific network leveraged the same input of 1,461 protein data and the concatenation was quite similar to the residual scheme; however, as we double checked the definition of residual network, we agreed that we misused it. Hence, we revised the name to Proteomic Neural Network (ProNNet) throughout the whole manuscript.

4. Github link does not work.

Response: We have updated the code website (line 475), please check out the following link: <https://github.com/jasonHKU0907/FutureHealthProteomicPrediction>

5. "It outputs of two outcomes, followed by linear activations, are the overall number of the 14 disease categories indexed before and after the baseline assessments." This sentence is unclear. Both the modeling and data aspects should be explained better. What does before and after baseline mean here? Does this mean that each patient can appear with different covariates for different outcomes?

Response: Thanks for the comments, to clarify the confusions, we enriched the description in Methods section (ProNNet and ProRS) in lines 377-391, which reads as:

“The comorbid network contains two identical branches with output targets as the total number of the 14 disease categories indexed before (left branch) and indexed after (right branch) the baseline assessments. In another word, the comorbid network was trained to learn how many disease categories have indexed in the past and will index in the future. Such information can somehow reflect the overall level of individual’s health conditions; thus, learnt latent features could donate potentially contribution to the endpoint-specific prediction task. The input of the comorbid network was 1,461 vectorized plasma proteins and it passed into two identical branches that contained four fully connected dense layers that each connected with *ReLU* activations of 512, 256, 128 and 64 nodes, respectively. As the number of disease category was a continuous variable, linear activation was adopted as the final activation function. ... The comorbid network was pre-trained, and its frozen weights of the last layers were passed into the endpoint-specified network.”

6. "Thus, the comorbid network is capable of overwhelmingly estimating the health conditions of each individual." Meaning unclear. Can this network predict the disease outcome with high accuracy on the test data?

Response: To clarify this confusion, we have reworded the descriptions in the previous response. The number of comorbidities (defined as using 14 disease categories in ICD-10 codes) can somehow reflect the overall level of individual’s health condition; thus, leant latent features could donate potentially contribution to the endpoint-specific prediction task. We performed leave-one-region-out cross-validation, and the comorbid network yielded mean absolute errors (MAE) of 1.84 ± 0.32 and 1.41 ± 0.10 for a number of comorbidities before and after the baseline, respectively. The comorbid network was an intermediate module within the ProNNet, which was not critical to our main results and key findings; thus we only briefly added a description in the Method section (ProNNet and ProRS) in lines 387-390 as follows:

“The performance of comorbid network yielded mean absolute errors (MAEs) of 1.84 ± 0.32 and 1.41 ± 0.10 for the number of comorbidities before and after the baseline, respectively, and fitted results were shown in Supplementary Fig. 7.”

Supplementary Fig. 7: Fitted scatterplot of comorbid network

Number of comorbidities was defined as the summed number of 14 disease categories listed in ICD-10 code.

a. Predicted versus actual number of comorbidities before baseline.

b. Predicted versus actual number of comorbidities after baseline.

7. "...with early stopping criteria that no validation loss after 25 epochs" Missing word here?

Response: Thanks for your comments, we reworded this sentence (lines 398-399) as follows:

“To avoid overfitting, the model training process early stopped upon the epoch that no decremental of validation loss observed after next 25 iterations.”

Reviewer #3 (Remarks to the Author):

The authors present a new index of disease risk called ProRS, based on proteomic measures, and show that it outperforms some commonly used predictors of various diseases in a subset of the UK BioBank data set. While I found the research presented in the paper impressive, I have a number of comments that I hope may improve the work. These are listed in roughly descending order of importance.

1. The authors point out that many existing predictive models in proteomics have been cross sectional, which clearly is not ideal for prediction. However, it is not clear to me how they are addressing this issue. Does the UK biobank have longitudinal measurements? If so, the authors should clarify when the proteomics measurements were taken relative to the baseline for their time-to-event analyses.

Response: We agreed that most existing proteome-based studies were performed using cross-sectional analysis. Several of them leveraged case-control approach to understand the plasma proteomic differences between healthy population and individuals diagnosed with certain diseases, e.g., dementia¹, Alzheimer’s disease², coronary heart disease³ and Type I diabetes⁴. Although these studies have been informative, they suffered from a lack of longitudinal measurements, largely limiting their application to real clinical utility. In the Introduction section (lines 72-80), we stated such limitation as follows:

“To date, most proteomic-based predictive studies were undertaken in cross-sectional manners, and several of them leveraged case-control approaches to understand the plasma proteomic difference between healthy population and individuals diagnosed with certain disease, e.g., dementia¹,

Alzheimer's disease², coronary heart disease³ and Type I diabetes⁴. Possibly biased by reverse causality, such studies have failed to identify proteomic signatures prior to disease onset. Although case-control studies are informative and can elaborate disease-related protein profiles to certain extent, longitudinal designs that estimate early molecular signatures associated with disease incidence are more appropriate for risk stratification.”

In our analysis, we extracted the clinical records from UK Biobank and excluded those participants who were previously diagnosed with specific disease categories to keep our study subjects as a relative healthier population. The participants' diagnosis information has been follow-up with a median 14.1 years after participants' baseline visits. We then utilized the proteomic data and other clinical related information collected at baseline to model their future incident of diseases or mortalities. We described this proteomic data and follow-up clinical diagnosis in the Methods section (UK-Biobank study cohort) as follows:

(lines 332-336): “The plasma proteomic data provided by the UKB was extracted from plasma assays collected at participants' baseline recruitment for a subset of 52,705 individuals, and those who had over 30% of missingness in proteomics data were excluded (n=699). We finally included 52,006 participants who had a median follow-up of 14.1 years until March 2023.”

(lines 341-350): “Endpoints were ascertained and classified according to the ICD 10 codes (Supplementary Table 1), ..., Participants' follow-up started from their baseline visits to the assessment center and was censored upon the earliest date of disease diagnosis, death, or the last available date from the hospital or general practitioner, whichever occurred first. Exclusion criteria to each endpoint was defined as any diagnosis sourced from self-reported clinical records or any incidents indexed before the baseline of the respective disease category that the endpoint belongs to.”

Reference:

1. Tanaka T, Lavery R, Varma V, Fantoni G, Colpo M, Thambisetty M, Candia J, Resnick SM, Bennett DA, Biancotto A. Plasma proteomic signatures predict dementia and cognitive impairment. *Alzheimer's & Dementia: Translational Research & Clinical Interventions*. 2020;6:e12018.
2. Hye A, Lynham S, Thambisetty M, Causevic M, Campbell J, Byers H, Hooper C, Rijdsdijk F, Tabrizi S, Banner S. Proteome-based plasma biomarkers for Alzheimer's disease. *Brain*. 2006;129:3042-3050.
3. Clarke R, Von Ende A, Schmidt LE, Yin X, Hill M, Hughes AD, Pechlaner R, Willeit J, Kiechl S, Watkins H. Apolipoprotein proteomics for residual lipid-related risk in coronary heart disease. *Circulation Research*. 2023;132:452-464.
4. Limonte CP, Valo E, Drel V, Natarajan L, Darshi M, Forsblom C, Henderson CM, Hoofnagle AN, Ju W, Kretzler M. Urinary proteomics identifies cathepsin D as a biomarker of rapid eGFR decline in type 1 diabetes. *Diabetes care*. 2022;45:1416-1427.

2. I would like to see more detail about the model testing process, maybe added to Figure 1 in the paper. It is not clear what segment of the data was held out for testing the predictive models, so a diagram would be helpful. Alternatively if the only validation was cross-validation, then the authors should be explicit about that.

Response: Thanks for your question, we did not perform evaluations in a hold-out validation set, both the ProNNet and following survival analysis were evaluated using leave-one-region-out cross validation. Following your advice, we drew a pipeline flowchart. Due to the limited space in Figure 1, we added it

as supplementary Fig 1. Additionally, as both you and Reviewer 2 concerned the details of model training and validation procedures, we enriched the description in the Method section (Statistical analysis) in lines 443-458. Please see our response to Reviewer 2 Question 2, where we attached the flowchart and revised descriptions.

3. I am concerned about the claimed Bonferroni correction applied to the Cox model results. Did the authors in fact adjust the p-values or did they just use a lower threshold of 0.01 on the raw p-values? If the former, then they should clarify what denominator they used and how they define a “family” of tests when controlling the type-1 error rate; if the latter, then this is surely not adequate to handle the multiplicity of tests performed.

Response: Sorry for the confusions due to our inaccurate expressions. We performed Bonferroni correction of the former approach as you mentioned in the comment. We defined the statistical significance as p-value less than the family-wise error rate (FWER) of $0.01/1,461$, where 1,461 was the number of protein family. We have amended this in the Methods section (Proteomic attribution estimates) in lines 425-427 and all relevant expressions in the manuscript, reads as:

“Bonferroni corrections was applied across all association hypothesis tests by using the family-wise error rate (FWER) of 6.84×10^{-6} ($0.01/1,461$) as the threshold of statistical significance.”

4. The statistical methods used to compare the performance of different predictive models are unclear. For example, line 139 states that ProRS had “significantly greater” performance. This is not a trivial problem as the metrics used are not simple sums of IID random variables, and comparing them additionally requires taking into account the correlation between metrics calculated on the same sample. The authors should provide more details about how this was accomplished.

Response: Thanks for pointing out this critical issue. As you mentioned, we previously treated the evaluation metrics as IID random variables, and employed Student t-test for comparison of significance. We did agree with your insightful comments, and reperformed statistical test by using approach previously developed by Kang et. al. Given censored time and ground truth from the same population, the approach was designed to test the significance between two correlated C-index and the method was implemented through R package *CompareC* (v1.3.2)¹. We have updated all results (mainly in Supplementary Table 7) and relevant descriptions in both Statistical Methods and Results in the manuscripts as follows:

(Methods section, lines 409-412): “Statistical significance test between paired C-index was implemented through a one-shot nonparametric approach in consideration between metrics calculated on the same sample, and this was implemented through R package *CompareC* (v1.3.2)¹.”

(Results section, lines 148-157): “In most endpoints, ProRS alone had significantly greater or comparable discriminative performance than Age+Sex, Serum, and PANEL. Furthermore, the ProRS significantly outperformed all three sets of clinical predictors in particular diseases, including five disease categories (diseases of infections, blood and immune disorders, nervous system disorders, respiratory system disorders, and genitourinary system disorders), seven specific diseases (bacterial and viral infections, leukaemia, anemia, dementia, heart failure, and chronic obstructive pulmonary disease), and all-cause mortality and its four causes (Fig. 3, Supplementary Table 7).”

(Results section, lines 158-166): “When incorporating ProRS into the Age+Sex or Serum models, a significant enhancement in predictive capability was observed across almost all endpoints (13 disease categories, all-cause mortality, 23 specific diseases, and four causes of death in the Age+Sex model; all

endpoints in the Serum model), but the combination did not significantly exceed ProRS alone in most endpoints. Of note, the protein-only model exhibited significantly improved discrimination in predicting breast cancer, prostate cancer, leukemia, dementia, Parkinson's disease, all-cause mortality, death caused by the nervous system, death caused by the circulatory system, and death caused by the respiratory system when compared to the combination of Serum and ProRS.”

(Results section, lines 167-173): “Adding ProRS to PANEL significantly improved predictive information over PANEL for 11 disease categories, all-cause mortality, 20 specific diseases, and four causes of death. It’s worth noting that, in more than one third of endpoints, the combination of ProRS and PANEL produced comparable C-indexes to ProRS alone. For the remaining endpoints, combining PANEL with ProRS yielded improved prediction performance compared to models based solely on single domain source data. However, the extent of the enhancement in predictive capabilities was limited when compared to using ProRS alone.”

Reference:

1. Kang L, Chen W, Petrick NA, Gallas BD. Comparing two correlated C indices with right-censored survival outcome: a one-shot nonparametric approach. *Statistics in medicine*. 2015;34:685-703.

5. The authors describe using area under the ROC curve (lines 410-411) but they apparently distinguish this from the c-index. My understanding was that the c-index (or c-statistic) was just another name for AUC. Do the authors mean something else when they use c-index?

Response: Thanks for your question, the Harrell’s concordance index (C-index) is a generalization of the area under the Receiver Operating Characteristic (ROC) curve (AUC), and they are both used to measure the discrimination power of prediction model. C-index can further take into account censored data; thus, it is commonly used in survival analysis. In our analysis, C-index was used when evaluating performance of Cox proportional hazard regressions where it fitted censored survival data. As for classification models to binary outcomes, AUC was used.

Regarding this question (lines 410-411), evaluations on different incident time windows, e.g., within 5-year, 10-year and beyond 10-year time window, the target outcomes were binary variables, and they were defined as either 1 or 0, representing any disease incident or not incident within specific time period after baseline assessment.

Reviewers' Comments:

Reviewer #1:

Remarks to the Author:

The authors have properly addressed all my concerns.

Reviewer #2:

Remarks to the Author:

I want to thank the authors for their careful revision and willingness to clarify the paper according to the reviewers' concerns.

I am especially happy about the new chart showing the cross validation approach which was my biggest concern.

I recommend the paper for publication.

Minor comment: The Github repo is very sparse as it stands now. Please add more information on how to use.

Reviewer #3:

Remarks to the Author:

The authors have satisfactorily addressed nearly all my comments. I am still unsure of how they defined time 0 (baseline) for the time-to-event analyses but probably this could be fixed with a sentence or two.

Responses to the reviewers of Nature Communications paper NCOMMS-23-26906A.

We would like to express our sincere gratitude for your constructive comments and suggestions on our manuscript. We have carefully considered each of your comments and made efforts to revise the manuscript accordingly. Here, we provide point-by-point responses to your comments and explain the changes we have made to the manuscript. **Our responses below are preceded by --, and the changes made to the paper are indicated below within "...", and in the revised paper in red font.**

Detailed response to reviewers

Response to Reviewer #1:

Reviewer #1 (Remarks to the Author):

The authors have properly addressed all my concerns.

Response to Reviewer #2:

Reviewer #2 (Remarks to the Author):

I want to thank the authors for their careful revision and willingness to clarify the paper according to the reviewers' concerns.

I am especially happy about the new chart showing the cross validation approach which was my biggest concern.

I recommend the paper for publication.

Minor comment: The Github repo is very sparse as it stands now. Please add more information on how to use.

Response: Thanks for pointing it out, we have enriched relevant documentations on how to implement model training and evaluation of the code.

Response to Reviewer #3:

Reviewer #3 (Remarks to the Author):

The authors have satisfactorily addressed nearly all my comments. I am still unsure of how they defined time 0 (baseline) for the time-to-event analyses but probably this could be fixed with a sentence or two.

Response: Thanks for this critical suggestions. The baseline time was set as participants' first visit upon the UK Biobank assessment centers, where blood samples were collected, stored, and subsequently analysed to obtain the proteomic data used for our study. Thus, the time-to-event analyses were performed using baseline collected data (proteomic profiles and clinical Panel) to establish prediction models for future incident diseases.

--To better illustrate this critical point, we amended the descriptions in the Method section (UK-Biobank study cohort) in lines 346-350 as follows:

“Participants’ follow-up started from the date of their first visits to the UKB assessment centers (baseline time), the same time that blood samples and other clinical information were collected, and participants’ follow-up were censored upon the earliest date of disease diagnosis, death, or the last available date from the hospital or general practitioner, whichever occurred first (censored time).”